# Bioadhesive Nanoparticles for Local Drug Delivery

**DOI:** 10.3390/ijms23042370

**Published:** 2022-02-21

**Authors:** Liu Yu, Zewen Luo, Tian Chen, Yaqi Ouyang, Lingyun Xiao, Shu Liang, Zhangwen Peng, Yang Liu, Yang Deng

**Affiliations:** School of Pharmaceutical Sciences (Shenzhen), Sun Yat-sen University, Guangzhou 510275, China; yuliu@mail2.sysu.edu.cn (L.Y.); luozw23@mail2.sysu.edu.cn (Z.L.); chent293@mail2.sysu.edu.cn (T.C.); ouyyq3@mail2.sysu.edu.cn (Y.O.); xiaoly37@mail.sysu.edu.cn (L.X.); liangsh59@mail2.sysu.edu.cn (S.L.); pengzhw8@mail2.sysu.edu.cn (Z.P.)

**Keywords:** bioadhesion, nanoparticles, bioadhesive polymer, local drug delivery

## Abstract

Local drug delivery is an effective strategy for achieving direct and instant therapeutic effects. Current clinical treatments have fallen short and are limited by traditional technologies. Bioadhesive nanoparticles (NPs), however, may be a promising carrier for optimized local drug delivery, offering prolonged drug retention time and steadily maintained therapeutic concentrations. In addition, the possibility of clinical applications of this platform are abundant, as most polymers used for bioadhesion are both biodegradable and biocompatible. This review highlights the major advances in the investigations of polymer-based bioadhesive nanoparticles and their innumerable applications in local drug delivery.

## 1. Introduction

Although systemic drug delivery is an advantageous delivery route due to its accelerated onset of action, it is often difficult to maintain the required local drug concentration. Simply administering higher dosages is often not a feasible solution coupled with the restriction of limited extravasation from the bloodstream into the target sites. These factors can lead to higher toxic effect and adverse side effects. Local drug delivery, however, aims to provide an optimal therapeutic effect directly to the disease sites with minimal or no systemic toxicity [1]. An excellent local drug delivery platform should be able to release the active pharmaceutical ingredient at a suitable and continuous rate, maintain efficacy, and reduce or eliminate potential adverse reactions. In order for the above functions to be achieved, various formulations have been developed for local drug delivery such as patch, spray, or micro- or nano-carriers [2]. All strategies aforementioned can be directly applied to target sites like the oral cavity or skin [3,4]. Some limitations are apparent before progression into clinical applications, including the stability and maintenance of drug concentrations following application.

The concept of bioadhesion refers to the interactive forces between the biological or synthetic material and a mucosal surface [5]. Specifically, bioadhesion is the interaction and/or chemical bonds between the polymer and a biological substrate, such as oral mucus, nasal mucus, or skin [6]. Bioadhesive nanoparticles have been explored for local drug delivery action and can be divided into natural biopolymer-based and synthetic or semi-synthetic polymer-based [7,8]. Generally speaking, natural biopolymers such as chitosan, gelatin, and lectin are biocompatible and can provide bioadhesive interactions. Synthetic or semi-synthetic polymer-based systems, however, deliver optimal adhesion in comparison to natural biopolymers but may induce increased chronic inflammation or higher cytotoxicity from potentially toxic degradation products (Figure 1) [7,8]. However, bioadhesion is not free of drawbacks, mainly due to possible cell toxicity, as well as weak tissue adhesive strength [7]. The nanotechnology approach through nanoparticles (NPs), however, may overcome limitations in bioadhesion.

Bioadhesive NPs can not only prolong drug retention time but can also encourage particle uptake and enhance local drug delivery with the combination of their small size and high specific surface area [9,10]. This review highlights the essential characteristics and applications of bioadhesive NPs in local drug delivery. In particular, this work focuses on the key polymers that exhibit bioadhesive properties, its related applications, and the prospects of bioadhesive NPs for local drug delivery.

## 2. Mechanism of Bioadhesion

“Bioadhesion” is the binding between natural or synthetic biopolymers and mucosa or cell surfaces [11]. When biopolymers adhere to a cell surface, the term “cytoadhesion” is typically used, whilst “mucoadhesion” is the bioadhesion to the mucus membrane [12,13]. Cytoadhesion is the direct covalent or non-covalent binding between bioadhesive polymers and cell surface components such as receptors or proteins [14,15]. In comparison, the process of mucoadhesion is more complex and can be described in three stages: (I) contact stage, (II) interpenetration stage, and (III) consolidation stage, as shown in Figure 2 [16]. The contact stage is initiated by the wetting of the mucoadhesive polymers to form close interaction between the polymers and mucosal surface [17]. The hydration state of the adhesive material affects the contact process, and the spreading of polymer solvents will increase the interaction region [18]. Afterwards, the chains of bioadhesive polymers penetrate the mucin glycoproteins, which results in chain entanglement during the interpenetration stage [16]. The consolidation stage involves chemical (covalent bonding, hydrogen bonding) and mechanical interactions (physical entanglement between polymers and mucin chains) between the polymer and mucosal surface, further strengthening the mucoadhesive binding force started in the interpenetration stage [16,19].

## 3. Materials and Related Applications of Bioadhesive NPs

### 3.1. Natural Biopolymer-Based Bioadhesive NPs

#### 3.1.1. Chitosan-Based Bioadhesive NPs

Chitosan, a type of linear polysaccharide, consists of randomly distributed β-linked D-glucosamines and N-acetyl-D-glucosamines. It can be easily extracted and prepared from shrimp and crab shells [20]. It has been extensively studied and applied in various fields, especially in drug delivery systems due to its biocompatibility, degradability, solubility, and non-toxicity. Chitosan also exhibits superior adhesion as its amino and carboxyl groups can form hydrogen bonds through the interaction of lipoproteins with the cell membrane [21]. The chain flexibility, strong electrostatic interaction, and surface energy properties of chitosan may also contribute to its adhesion property [22]. On the basis of these features, researchers have designed many chitosan-based bioadhesive delivery systems to enhance contact with cell surfaces and extend residence time, thereby promoting drug absorption and strengthening the locally delivered drug’s therapeutic effect. In a recent study, Han et al. designed orally deliverable nanoparticles with the self-assembly of chitosan and water-insoluble drugs curcumin (CUR) or 7-ethyl-10-hydroxycamptothecin (SN38) for the synergistic treatment of inflammatory bowel diseases (IBDs) and colitis-associated colorectal cancer (CAC) (Figure 3) [23]. In addition to being the primary carrier for drug delivery, chitosan can also render an adhesion property to non-adhesive delivery systems as a surface coating. Cordenonsi et al. developed chitosan-coated nanostructured lipidic carriers (NLCs) loaded with fucoxanthin (FUCO) to effectively inhibit excessive skin proliferation while maintaining skin integrity in psoriatic skin [24]. Here, the chitosan coating could facilitate interaction with the target through biological adhesion so that the therapeutic effect of the drug can be enhanced.

#### 3.1.2. Gelatin-Coated Bioadhesive NPs

Gelatin is a generic name for partially acidic or alkaline hydrolysates of collagen from animals [25]. It is generally considered as one of the most promising biomaterials because of its complete biodegradability, biocompatibility, and unique biological properties. The carboxyl, amine, and hydroxyl groups in the molecular structure of gelatin form a multitude of hydrogen bond groups that contribute towards high hydrophilicity, swelling rate, and electrostatic interaction [16]. However, the mechanical and adhesive properties of gelatin are inadequate, limiting its clinical application as an end-use material. Researchers then proposed a series of strategies to improve its bioadhesion through either chemical modification such as thiolation, meth-acrylation, catechol conjugation, and amination, to name a few, or blending gelatin with other biomaterials [16]. Montazerian et al. constructed a novel gelatin type- and methacryloyl-based hydrogel with dopamine, which resulted in superior adhesion performance and robust mechanical properties. The hydrogel may provide a potential choice for skin-attachable substrates [26]. Barron et al. used gallic acid (GA, L-DOPA analogue) to redesign gelatin with efficient bioadhesive properties. It has been shown that gelatin–gallic acid/ZnO nanocomposites have enhanced adhesion and antibacterial properties, which could be of benefit in its application in wound and burn dressing materials [27].

#### 3.1.3. Dopamine-Based Bioadhesive NPs

As is widely known, marine mussels can maintain outstanding clinging properties under extremely harsh marine environments. Recent research has proposed the use of L-3,4-dihydroxyphenylalanine (DOPA) as a bioadhesive functional coating of nanoparticles. DOPA is present in the mussel adhesive protein at high concentrations, with the ortho-dihydroxyphenyl (catechol) functional group. This functional group can form strong bonds with a mass of inorganic or organic surfaces in an aqueous environment [28,29,30,31]. A study by Carvalho et al. showed the development and application of a multilayer film that was based on marine mussels’ adhesion properties. In combination with integrating bioactivity of bioglass nanoparticles and silver for its antibacterial properties, this technology may have prospects in orthopedic implants [29]. Derivatives of dopamine such as poly(dopamine) (DP) or nitrodopamine (NDP) have a similar function. DP and NDP coatings were found to enhance the cell adhesion and biocompatibility of TiO2 nanotubes. In addition, NDP-coated TiO2 nanotubes showed enhanced osteogenic potential compared with DP-coated substrates, indicating its potential for application in bone regeneration [32]. Other than the coating, Zhou et al. synthesized a nanocapsule with a series of bioadhesive and biocompatible polydopamine-grafted lignin (AL-PDA) through the free radical addition of dopamine (DA) and alkali lignin (AL) [30]. This product could be incorporated into sunscreen formulations for its antioxidant capacity and biocompatibility.

#### 3.1.4. Lectin-Based Bioadhesive NPs

Lectins are naturally occurring glycoproteins or sugar-binding proteins. Lectins can bind directly to the epithelial cells themselves, and therefore they have cytoadhesive properties [33]. This is of particular benefit in treating gastrointestinal diseases as it can overcome the limitation of mucus gel layer renewal times in comparison to non-specific adhesion of other nanomaterials. Moulari et al. constructed drug-loaded nanoparticles coated with peanut (PNA) and wheat germ (WGA) lectins for active targeting and selective adhesion to inflamed tissue in experimental colitis [34]. The study showed that due to the specific binding of PNA to its receptors in intestinal inflammatory tissue [35], the selectivity of biological adhesion to inflamed tissue was increased, which subsequently improved treatment efficacy. In addition, Mostaghaci et al. proposed a method to bind bacterial robots to epithelial cells that could allow for the deposition of mannose on the membrane. Specifically, targeted drug delivery could be achieved by utilizing the affinity between the lectin molecules at the hair tip of type I bacteria and mannose molecules of the epithelial cell membrane (Figure 4) [36].

#### 3.1.5. Alginate-Based Bioadhesive NPs

Sodium alginate is a by-product of the extraction of iodine and mannitol from brown algae kelp or *Sargassum*. As a polysaccharide consisting of abundant hydroxyl and carboxyl groups, it has natural adhesive properties, as well as excellent biocompatibility and degradability. Due to the innate advantages, the incorporation of alginate has been used widely in the fields of food and medicine. Fernandez et al. used sodium alginate to modify the nanostructured lipid carriers (NLCs) prepared by shea butter and argan oil. These bioadhesive NLCs loaded two antioxidants, and tea oil enhanced the retention of drugs on the skin, effectively releasing them into the wound to facilitate the healing process [33].

### 3.2. Synthetic or Semi-Synthetic Biopolymer-Based Bioadhesive NPs

Apart from natural biopolymers, synthetic biopolymers are also comprehensively applied to local drug delivery systems, owing to their potential of modification, versatility, and low batch-to-batch variation.

#### 3.2.1. Hyperbranched Polyglycerol-Coated Bioadhesive NPs

Polyethylene glycol (PEG) is ubiquitously used in NP coatings due to its low toxicity and its resistance to non-specific binding of biomolecules. PEG coatings could considerably prolong the circulation of blood of NPs. However, most intravenously injected PEG-coated NPs are still observed to accumulate in the liver [37,38]. In contrast, a novel form of NPs based on polylactic acid block–hyperbranched polyglycerol (PLA-HPG) copolymers also known as non-bioadhesive NPs (NNPs) exhibited prolonged circulation of blood time and less accumulation in the liver, which is expected to be superior to the PEG-coated NPs [39]. Copolymers of PLA-HPG were synthesized by a one-step esterifification, and NPs were produced by a single emulsion using PLA-HPG.

Furthermore, NNPs could be converted to bioadhesive NPs (BNPs) by the oxidation of surface vicinal diols into aldehydes with NaIO_4_ treatment. The aldehyde groups will form Schiff base bonds with amino groups, leading to bioadhesion on protein-rich surfaces (Figure 5). These BNPs can be universally applied for different types of tissues such as skin or inner mucosa. For instance, Deng et al. designed a novel sunscreen based on BNPs to prevent the penetration of reactive oxygen species produced after UV photochemical activation and subsequent reactive oxygen species-mediated DSBs, which produces an excellent anti-UV efficacy [14]. This finding may provide new insights for sunblock formulation design. Moreover, in a pilot clinical study, it was discovered that BNPs containing avobenzone and octocrylene created broad-spectrum ultra-violet radiation protection. The results demonstrated that the sunscreen formulation may provide higher safety and performance superiority [40]. This can potentially expand the concept of traditional sunscreens. Recently, Hu et al. applied the BNP drug delivery system loaded with camptothecin to the squamous cell carcinoma (SCC). The results significantly improved the chemotherapeutic efficiency by enhancing the local drug retention time [41]. Apart from skin applications, the drug delivery system can also be employed in the peritoneum and intravaginal membrane. Deng et al. utilized the nanoplatform to treat serous uterine carcinoma via intraperitoneal injection with the significantly extended retention time of the NPs in the peritoneal space, producing a potent chemotherapeutic effect and lower systemic toxicity [42]. It was shown that the BNPs could prolong the retention time of elvitegravir, an antiretroviral drug, through intravaginal delivery [43]. It is therefore worth exploring the delivery of BNPs on other diseases such as oral cancers or conjunctivitis, whereby such technology could overcome current formulation challenges. Meanwhile, although little research has been conducted on them, there will be an enormous development prospect for further research based on modified PLA-HPG or even BNPs such as ligands.

#### 3.2.2. Poly (Acrylic Acid)-Based Bioadhesive NPs

Poly (acrylic acid) (PAA) is generally regarded as safe by the U.S. Food and Drug Administration (FDA) when administrated orally, showing low toxicity and irritation [44]. Carbopol is a well-known commercially available co-polymer of PAA. Their bioadhesive ability may be attributed to the carboxylic acid groups forming hydrogen bonds with mucins on the mucosa [45]. Zou et al. incorporated PAA to develop novel bioadhesive NPs that (i) improved DNA-binding efficacy, (ii) protected DNA from enzymatic degradation in vitro, and (iii) exhibited higher transfection efficiency. The bioadhesive NPs may be a prospective candidate as a non-viral carrier for lung cancer gene therapy [46]. In addition, Vakili et al. established a PAA-grafted cellulose NP mucoadhesive hydrogel. The work showed it to be beneficial for the local delivery of cisplatin in terms of lower intrinsic cytotoxicity and in improving the IC_50_ in HCT116 cells, being advantageous in the development of a formulation for colorectal cancer treatment [47]. Recently, Khutoryanskaya et al. constructed mucoadhesive films based on NPs comprised of PAA and methylcellulose (MC). The presence of PAA helped improve bioadhesion of the formulation. For example, the in vivo retention and release of riboflavin in the films on corneal surfaces could be controlled according to the demand by changing the PAA/MC ratio. The films were proved to be lower-irritant to mucosal surfaces at pH = 4 and were retained at the administration site for 30–60 min. As such, the films could serve as a suitable platform for ocular drug delivery [48].

#### 3.2.3. Cellulose Derivative-Based Bioadhesive NPs

As the major constituent existing in plant cell walls, cellulose has been widely investigated for medical applications due to its biocompatibility. However, its main disadvantage is its poor processability, owing to its weak solubility in common organic solvents. Cellulose derivatives are feasible alternatives compared to pure cellulose [49]. Cellulose derivatives are indispensable pharmaceutical excipients with a wide range of applications [50]. Some cellulose derivatives, such as hydroxypropyl methyl cellulose phthalate, present mucoadhesive properties due to the presence of hydroxyl groups that can form hydrogen bonds with mucins [51]. Researchers have explored the design of bioadhesive NPs utilizing cellulose derivatives. For instance, Kovtun et al. prepared calcium phosphate NPs loaded with chlorhexidine. The study showed that NPs that were coated with carboxymethyl cellulose were successful in its bioadhesion to enamel and dentin. In addition, the bioadhesive NPs were also found to suppress bacterial growth [52]. Banlunara et al. demonstrated that the amount of *Helicobacter pylori* (*H. pylori)* in a C57BL/6 mouse model was significantly decreased following oral administration of ethyl cellulose (EC) NPs encapsulating clarithromycin. The superior treatment effect of encapsulated clarithomycin is due to the improved adherence of EC NPs to the stomach mucosa of treated mice, which may suggest a potential use of these NPs for the oral delivery treatment of *H. pylori* infections [53].

## 4. Test Models

There are a series of test models established for the characterization of bioadhesive properties. In in vitro models, Deng et al. utilized poly(L-lysine)-coated slides to stimulate the protein-rich tumor microenvironment and tested the performance of bioadhesive nanoparticles. They immobilized the surface of slides with unloaded NPs, EB-loaded NPs (EB/NNPs and EB/BNPs), free EB, or PBS and evaluated the in vitro efficacy in order to suppress the growth of USC cells. They observed that only slide regions pretreated with EB/BNPs significantly suppressed the growth of tumor cells [42]. In addition, Deng et al. smeared DID/NNPs and DID/BNPs on porcine skin, which is a compatible model for mimicking human skin. This was cultivated for 6 h in a humidity chamber at 32 °C, followed by washing with PBS. The study found that BNPs showed a significantly higher skin retention than NNPs [14]. In terms of ex vivo models, the bioadhesion of materials are mostly reflected by fluorescence intensity with respect to time. For example, Han et al. showed that the active material, Cy5.5-labeled nCUR, possessed a longer colonic retention time in colitis mice models than in healthy colonic mice tissue by ex vivo NIR imaging at 6 h and 24 h [23]. In in vivo models, researchers can monitor bioadhesion through live imaging. Deng et al. observed that orally administered IR-780/BNPs were retained for 5–10 d in comparison to intraperitoneal administration of IR-780/NNPs in mice through live imaging [42].

## 5. Current Application Directions

One of the primary purposes of designing and applying bioadhesive materials is to achieve slow and sustainable drug release at the target site by enhancing cargo retention [34]. It is known that there are various complex physiological barriers in the different segments of gastrointestinal tract that greatly restrict the absorption of drugs. Carriers with bioadhesive properties play important roles in local treatment of the gastrointestinal diseases and oral drug delivery systems by improving their cargo’s interaction with biological systems such as mucus barriers. Studies have shown that lectins can specifically target cells or tissues by binding to specific sugar groups. Hence, lectin-modified drug delivery systems may achieve further precise targeted delivery as second-generation bioadhesive materials. On the other hand, the skin and mucous membranes are the first barriers for human body protection. Combining bioadhesive materials with other delivery materials can achieve local drug delivery and sustain drug release on skin or mucous membranes. In the present, research on the applications of bioadhesive materials for skin diseases such as skin cancer, psoriasis, wound management, peritoneal metastatic cancer, eye diseases, nasal cavity administration, and vaginal administration are being widely conducted, showing superior drug delivery effect as opposed to single systems.

## 6. Discussions and Conclusions

This review summarizes the recent advances of bioadhesive NPs for local drug delivery, mainly introducing a variety of polymers exerting bioadhesive properties as well as their related local applications. In conclusion, the bioadhesive NPs possess dual advantages of being nanocarriers and bioadhesives, making this platform more suitable for local therapies. Meanwhile, novel and multifunctional delivery systems based on bioadhesion are rapidly being developed in combination with nanotechnologies. For instance, tissue adhesives show excellent antibacterial and hemostatic properties when combined with the metal NPs, among which silver and gold are the most studied. Nano-compounds encapsulating growth factors or genes, or those combined with stem cells, have attracted great attention and interest from researchers, potentially providing novel therapeutic strategies in the near future [10]. In addition, bioadhesive NPs may be further improved when combined with advanced therapies such as intelligent response or photothermal/photodynamic therapy. However, besides serving as delivery vehicles themselves, some of these bioadhesive materials can also be used as coatings to render adhesive properties to other carriers and improve their biocompatibility. Halloysite nanotubes (HNT), for example, are naturally occurring aluminosilicates with a hollow tubular structure, similar to carbon nanotubes [54]. This cavity structure and high adsorption properties endow it with the advantages of high drug loading and sustained drug release. In addition, HNT can adsorb or graft functional molecules on the inner and outer walls such as in DNA [55]. If bioadhesive materials such as chitosan or dopamine are modified on the surface of HNT, the advantages of their adhesion and nanotubes can be combined to obtain a multifunctional delivery system. This combination approach can increase prospects in the application of local disease treatments for the superiority of slower release of drugs and longer drug retention [55,56,57]. Although significant developments have been made for bioadhesive NPs, significant barriers still exist concerning the nanomaterials’ long-term safety, anticipated behavior, and toxicity to the human body, among other factors, yet there are still many challenges and barriers existing. For example, they generally cannot exhibit the properties of long-term adhesion and lack more accurate delivery after local adhesion, limiting future applications [9]. Therefore, comprehensive research and clinical trials are expected to further optimize the understanding of bioadhesive NPs. Overall, the rapid advancements in bioadhesive NPs in pre-clinical research has illustrated the enormous potential for local drug delivery and will further promote the clinical application for disease treatments locally and specifically.

## Figures and Tables

**Figure 1 ijms-23-02370-f001:**
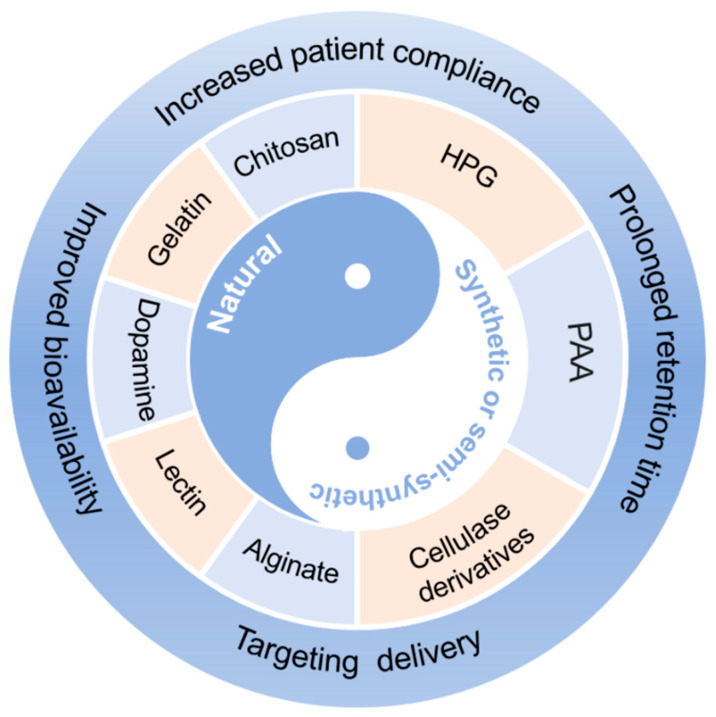
Bioadhesive nanoparticles (NPs) based on different materials. HPG: hyperbranched polyglycerol; PAA: poly (acrylic acid).

**Figure 2 ijms-23-02370-f002:**
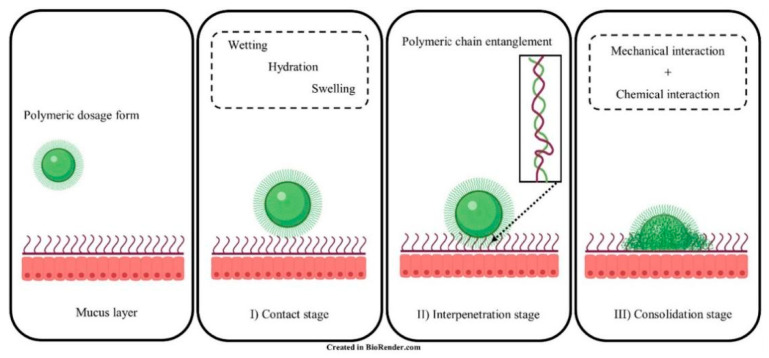
Illustration of mucoadhesive interaction(s) between polymers and mucin glycoproteins. I) Contact stage, II) Interpenetration stage and III) Consolidation stage. Adapted from Ahmady et al. [16].

**Figure 3 ijms-23-02370-f003:**
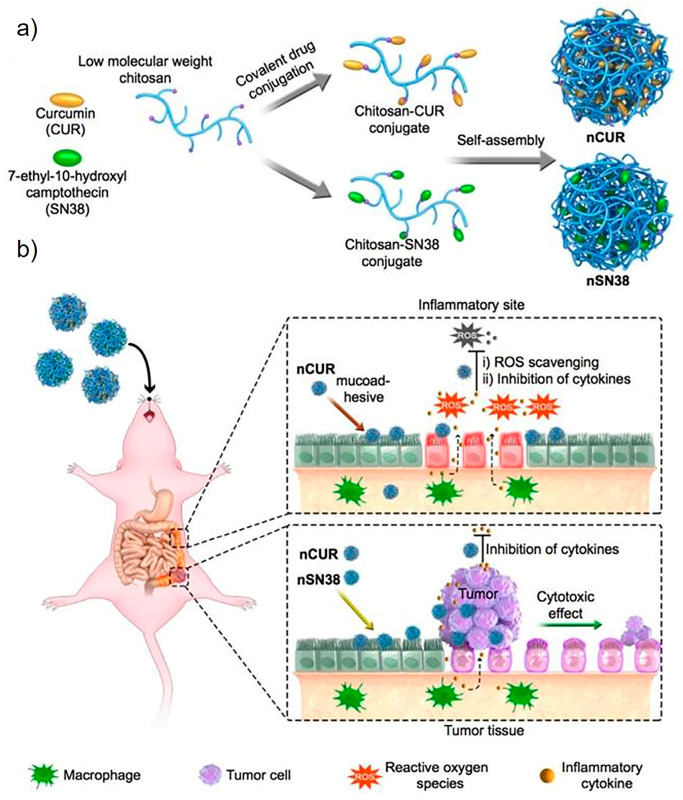
Schematic diagram of chitosan–drug conjugates self-assembling to form nanotherapeutic agents (i.e., nCUR and nSN38) and oral nanotherapeutic agents for CAC treatment. (**a**) SN38 and CUR therapeutics are separately attached to carboxylated chitosan by hydrolyzable bonds. The formed chitosan–drug conjugates self-assemble into stable colloids and bioadhesive nanotherapeutic agents that can be used for oral administration. (**b**) After water that contained the therapeutics ad libitum is drank to deliver the drug to the intestine, the nCUR and nSN38 nanotherapeutics are able to tightly adhere to intestinal villi and efficiently accumulate in the rodent’s inflamed colon tissues and tumors. Subsequently, intestinal inflammation and tumor growth are, respectively, suppressed with the gradual release of therapeutic drug components CUR and SN38. CUR: curcumin; SN38: 7-ethyl-10-hydroxycamptothecin. Adapted from Han et al. [23].

**Figure 4 ijms-23-02370-f004:**
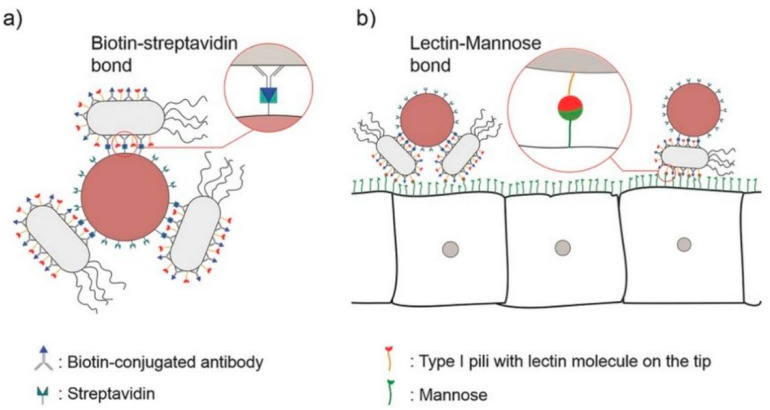
(**a**) A schematic of “bacterial robots”, i.e., synthetic particles combined with bacterial cells by biotin–streptavidin bonds. (**b**) Bacterial robots with lectin molecules at the tip of type I bacterial pili can attach to the disease site cells expressing mannose via lectin–mannose bonds. Adapted from Mostaghaci et al. [36].

**Figure 5 ijms-23-02370-f005:**
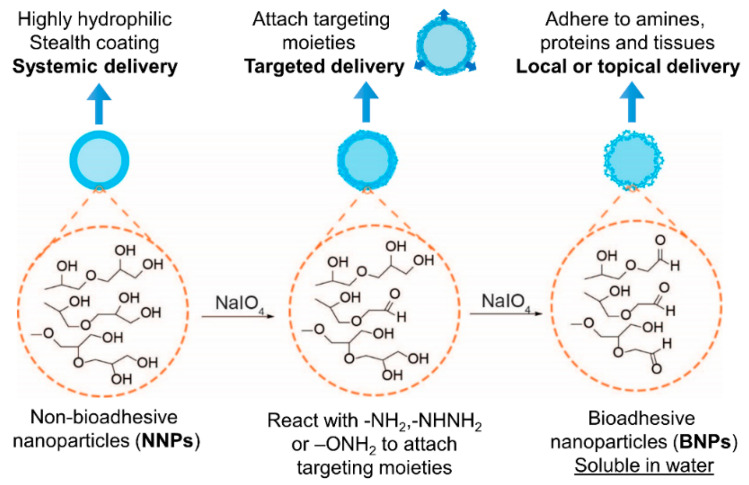
Synthetic schematic of the conversion of bioadhesive nanoparticles (BNPs) from non-bioadhesive nanoparticles (NNPs).

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
