# Peer review of "Bioadhesive Nanoparticles for Local Drug Delivery"

_ijms, 2022, doi:10.3390/ijms23042370_

Round 1

Reviewer 1 Report

This review describes the types and properties of bioadhesive NPs and their application in local drug delivery. The topic is focused and would be of interest to a broad readership. It provides a great introduction to other researchers not exactly in the research field, including the reviewer. Before publication, I recommend a thorough edit of the manuscript to improve the English, particularly in the Abstract and Introduction, which are particularly hard to read (e.g., should be systemic delivery rather than systematic delivery; detailly; targeting delivery; The Schematic diagram).

Other points to be addressed by the authors:

  1. “In order to achieve the above functions, various formulations have been developed for local drug delivery, such as tablets, capsules, micro- or nano-carriers, all of which can be directly delivered or released to target sites.” How are capsules and tablets directly delivered to the target site? What target site is referred to here?
  2. “In general, the spreading of fluid dosage forms will increase the interaction region”. What is meant by fluid dosage forms? This needs rephrasing.
  3. “Because of this ability to specifically bind to sugar groups, they played a variety of roles in biometrics, referring to cells, carbohydrates, and proteins.” Rephrase to clarify.
  4. “Sodium alginate is a by-product of iodine and mannitol”. May need to rephrase.
  5. “Although great developments have been made for bioadhesive NPs, considering nanomaterials’ long-term safety and anticipated behavior, toxicity to the human body, and so on, there are still many challenges and barriers existing”. Provide examples of the challenges and barriers that need to be overcome.
  6. When referring to references in the text, Surname et al. is sufficient (first name initial not needed).
  7. A good section that could be added would on the models used for testing the performance of bioactive materials from in vitro models to in vivo if applicable. Otherwise, the authors should refer to published work or reviews that might discuss test models in greater detail.

Reviewer 2 Report

The manuscript entitled “Bioadhesive Nanoparticles for Local Drug Delivery” describes the effective drug delivery strategy for achieving therapeutic effects.
The manuscript can be accepted after significant revision.
Authors need to change the style of whole text.

1) There is no uniformity in the alignment of captions to figures in width. Figures 1 and 3 are signed across the entire width of the page, while Figures 2, 4 and 5 are signed similarly to the main text of the article.

2) Line 13. The word “delivery” occurs twice in the sentence. It might be appropriate to lighten the sentence or paraphrase.

3) Line 15.  Perhaps it should be written as “In addition, the possibility of clinical applications of this platform is increasing…”

4) Line 17. Perhaps it should be written as “This review highlights the major advances in the investigations of bioadhesive nanoparticles…”.

5) Line 22. Probably “systemic” should be written instead of “systematic”.

6) Line 25. Probably “toxic effect” should be written instead of “toxicity”.

7) Line 27. Probably “systemic” should be written instead of “systematic”.

8) Line 27. Probably “because of this advantage” should be written instead of “because of this difference”.

9) Line 33. Probably “including the solving of the question how to effectively keep the drug concentration stable after application” should be written instead of “including how to effectively keep the drug concentration stable after application”.

10) Line 36. Probably “novel approaches for clinical applications, using of bioadhesive nanoparticles (NPs) being one of them.” should be written instead of “novel approaches for clinical applications, bioadhesive nanoparticles (NPs) being one of them”.

11) Lines 44-46. Probably “Generally speaking, natural biodegradable biopolymers  such as chitosan, gelatin, and lectin have outstanding biocompatibility and can provide bioadhesive interactions.” should be written instead of “Generally speaking, natural biopolymers such as chitosan, gelatin, and lectin can provide biodegradable bioadhesive interactions with outstanding biocompatibility.”.

12) Line 124. Probably “limiting its clinical application as an end-use material” should be written instead of “limiting its end-use and clinical application”.

13) Lines 136-137. Probably “As widely known, marine mussels can keep outstanding clinging properties under extremely harsh marine environments, so” should be written instead of “Inspired by marine mussels that can keep outstanding adhesive properties under extremely harsh marine environments”.

14) Lines 143-147. A difficult sentence to understand, obviously it would be worth simplifying or rephrasing.

15) Line 154. Probably “biomarking” should be written instead of “biometrics”.

16) Line 156. Probably “Lectins can bind directly to the epithelial cells themselves, so they have cytoadhesive properties” should be written instead of “Lectins can bind directly to the epithelial cells themselves, namely cytoadhesion”.

17) Lines 156-158. A difficult sentence to understand, obviously it would be worth simplifying or rephrasing.

18) Line 175. Perhaps it should be written as “Sodium alginate is a by-product of extraction of iodine and mannitol...”

19) Line 180-181. The word “antioxidant” occurs twice in the sentence. It might be appropriate to lighten the sentence by using the word "them” or paraphrase.

20) Lines 189-190. Probably “circulation of blood” should be written instead of “blood circulation”.

21) Line 193. Probably “circulation of blood” should be written instead of “blood circulation”.

22) Line 197. Probably “amino groups” should be written instead of “amino”.

23) Line 198. Probably “types of tissues” should be written instead of “occasions”.

24) Line 220. Probably “Generally recognized as safe” should be written instead of “safe”.

25) Line 221. Probably “low” should be written instead of “lower”.

26) Line 261. A difficult sentence to understand, obviously it would be worth simplifying or rephrasing.

Also it is necessary to expand the manuscript by including more modern works on the study of nanotubes, for example: doi.org/10.1016/j.clay.2021.106041, doi.org/10.3390/molecules25153557, doi.org/10.1016/j.clay.2020.105707 etc.

Reviewer 3 Report

Good material according to bioadhesive nanoparticles, but one can add it to the view of the other types of bioadhesive nanoparticles, for example, modified PLA-HPG for bioadhesia and other missing in this review. And also in more detail to submit the further development of the BNP. 

Round 2

Reviewer 2 Report

I thank the authors for the attention they paid to my edits.